# Multidisciplinary Clinical Study on Retinal, Circulatory, and Respiratory Damage in Smoking-Dependent Subjects

**DOI:** 10.3390/medicina61020347

**Published:** 2025-02-16

**Authors:** Marcella Nebbioso, Annarita Vestri, Magda Gharbiya, Mattia D’Andrea, Matteo Calbucci, Federico Pasqualotto, Serena Esposito, Alessandra D’Amico, Valentina Castellani, Sandra Cinzia Carlesimo, Paolo Giuseppe Limoli, Alessandro Lambiase

**Affiliations:** 1Department of Sense Organs, Sapienza University of Rome, Piazz.le A. Moro 5, 00185 Rome, Italy; magda.gharbiya@uniroma1.it (M.G.); mattia.dandrea@uniroma1.it (M.D.); matteo.calbucci@gmail.com (M.C.); sandracinzia.carlesimo@uniroma1.it (S.C.C.); alessandro.lambiase@uniroma1.it (A.L.); 2Department of Public Health and Infectious Disease, Sapienza University of Rome, Piazz.le A. Moro 5, 00185 Rome, Italy; annarita.vestri@uniroma1.it; 3Antismoking Center UOC Pneumology, I Clinical Medicine, Sapienza University of Rome, Viale del Policlinico 155, 00161 Rome, Italy; federico.pasqualotto33@gmail.com (F.P.); serenaesposito109@gmail.com (S.E.); 4Department of Medical-Surgical Sciences and Biotechnologies, Sapienza University of Rome, Corso della Repubblica, 04100 Latina, Italy; alessandra.damico@uniroma1.it; 5Department of Clinical Internal, Anaesthesiological and Cardiovascular Sciences, Sapienza University of Rome, Viale del Policlinico 155, 00161 Rome, Italy; valentina.castellani@uniroma1.it; 6Low Vision Research Centre of Milan, p.zza Sempione 3, 20145 Milan, Italy; paololimoli@libero.it

**Keywords:** optical coherence tomography angiography (OCTA), macular vascular density (FD), oxidative stress, smoking, spirometry, total thrombus formation analysis system (T-TAS), expiratory volume

## Abstract

*Background and Objectives*: Cigarette smoking is a widely prevalent risk factor in the global population, despite its well-recognized systemic impact. In this pilot study, an association was hypothesized between alterations in hemorheological and respiratory characteristics and damage at the chorioretinal level, considering that traditional cigarette smoking may increase oxidative stress, platelet activation, and thrombotic phenomena at the systemic level. Fundoscopy can provide information about the characteristics of the cerebral district and the entire circulatory system. Therefore, the aim of this research was to evaluate the impact of cigarette smoking on chorioretinal vascularization and pulmonary and blood parameters through investigations with optical coherence tomography angiography (OCTA), spirometry, and the total thrombus formation analysis system (T-TAS). *Materials and Methods*: Thirty subjects were recruited, divided into 20 traditional cigarette smokers (SMs) and 10 non-SMs, who underwent a comprehensive ocular examination, including OCTA. Spirometric evaluation and blood sampling were also performed on both groups to study pulmonary functional capacity, as well as T-TAS. *Results*: An analysis of the obtained data confirmed the systemic impact of smoking, evidenced by an increase in T-TAS and a decrease in forced expiratory volume in 1 s expressed in liters (FEV1 L) in SMs compared to the non-SMs group. Additionally, OCTA revealed a statistically significant alteration in macular vascular density (FD) in the right eye (RE) of the examined SMs. The other parameters evaluated did not show statistically significant differences. *Conclusions*: It is believed that FD, FEV1, and T-TAS may be promising values in correlating the alterations observed in SMs, as highlighted by the changes detected with OCTA, spirometry, and hemorheological data. Further research is needed to confirm and expand the results already obtained and to evaluate the systemic vascular damage and oxidative stress caused by tobacco consumption.

## 1. Introduction

Smoking is highly prevalent in the population, and although it is unequivocally recognized as harmful to health, millions of individuals continue to sustain this chronic and compulsive addiction, which has long been associated with the onset of numerous diseases. In Western societies, it is estimated that more than 10% of cardiovascular deaths are attributable to cigarette smoking due to chronic inflammatory progression, induced atherosclerosis, and oxidative stress damage to the vascular endothelium. Smoking can take various forms, but the most common is the use of cigarettes, which contain a combination of tobacco leaves and various additives, including flavorings, nicotine regulators, and combustion aids, not to mention structural components such as the paper used for assembly, printing ink, and finishing adhesives. The smoke contains more than 7000 different chemicals, 86 of which are identified to be carcinogenic. Over 100 of these are toxic substances responsible for the pathogenesis and progression of smoke-induced diseases [1,2], including solid and liquid particles such as nicotine, carbon monoxide, ammonia, acrolein, formaldehyde, acetone, hydroquinone, nitrogen oxide, cadmium, and tar. In addition to the well-known mutagenic and carcinogenic properties, other severe effects of smoking are related to increased oxidative stress and platelet activation. Cigarette smoking compromises the anti-inflammatory, anti-aggregating, and vasodilatory properties of the vascular endothelium. Consequently, molecules released by damaged cells contribute to the induction of immunity, coagulation, and inflammation [3,4,5].

In smokers (SMs), an increase in neutrophils, lymphocytes, and monocytes has been observed, along with elevated serum levels of inflammatory cytokines such as tumor necrosis factor-α (TNF-α), interleukin (IL)-1β, IL-6, and IL-8 [6,7], as well as the expression of adhesion molecules on the surfaces of endothelial cells (PECAM) [8].

These and other factors contribute to the progression of atherosclerosis. Furthermore, various cells within atherosclerotic lesions, including endothelial cells, macrophages, and dendritic cells, express Toll-like receptors (TLRs) [5,9]. In mouse models, TLR9 has been shown to play a key role in the development of atherosclerosis [10,11].

This has organic implications, particularly affecting three major systems:Respiratory system—development of asthma and chronic obstructive pulmonary disease (COPD) due to endothelial damage in pulmonary vessels;Cardiovascular system—increased risk of arterial hypertension (AHY), stroke, coronary artery disease, peripheral vascular diseases, and aortic aneurysms;Reproductive system—difficulties in conception and an increased frequency of miscarriage.

Chronic exposure to cigarette smoke leads to endothelial dysfunction, as demonstrated by the research of Celermajer et al., which shows a dose-dependent and time-dependent reduction in flow-mediated vasodilation [12,13].

Therefore, endothelial dysfunction, inflammation, and atherothrombotic effects induced by tobacco smoke contribute to negative impacts, particularly on the cardiovascular system, and can establish a vicious cycle of reciprocal damage. It has been reported that 75% of cases of sudden cardiac death due to acute atherothrombosis involved individuals with tobacco dependence [14,15].

The negative effects of cigarette smoking on the ocular system have recently been demonstrated, particularly using optical coherence tomography angiography (OCTA), a relatively new technique capable of assessing tissue structure and blood flow. Several studies conducted by various authors have thus investigated the potential impact on the morphological and hemodynamic characteristics of the central choroidal–retinal vasculature, although with results that are not entirely consistent. These studies have shown a significant reduction in the vascular flow index in smokers compared to the control group, along with ischemic effects, as well as a decrease in choroidal thickness in the macular area, both in the acute and chronic phases of smoking [16,17,18,19].

Based on these data, we assessed organic alterations in active SMs through diagnostic investigations in the pulmonary, hematological, and ocular fields. This pilot study aimed to evaluate the retinal impact of cigarette smoking, correlating parameters obtained through OCTA and hemorheological and spirometric values in subjects with nicotine dependence and non-SMs. The primary endpoint was to identify an OCTA parameter that could be correlated with retinal damage caused by smoking in individuals with tobacco dependence.

## 2. Materials and Methods

### 2.1. Participants

Thirty Caucasian Italian subjects were enrolled in this study, including 20 traditional cigarette SMs and 10 non-SMs. The research was conducted in collaboration with the Smoking Cessation Unit within the Department of Public Health and Infectious Diseases, the Department of Clinical Internal, Anesthesiological and Cardiovascular Sciences, and the Ocular Electrophysiology and Biomedical Techniques Center of the Umberto I Polyclinic, Sapienza University of Rome. Written informed consent was obtained from all participants. The study was approved by the Ethics Committee of Sapienza University of Rome (Ref. 3241 Prot. 549/17, date 12 June 2017) and was conducted in accordance with the Declaration of Helsinki.

The 30 individuals were enrolled after a detailed clinical medical history and included in this study only if they met the following inclusion criteria: age between 40 and 70 years, good general health, normal cardiovascular and respiratory compensation, normal body weight, normal intraocular pressure (IOP), no history of surgery on the lens or retina, and visual acuity with refractive errors and spherical equivalent no greater than ±3 diopters. The SMs were required to have been smoking for at least 10 years, and the non-SMs had never smoked. In the SM group, we assessed the number of daily cigarettes, years of smoking, and pack/years to enable a more straightforward quantitative evaluation of the smoking intensity for everyone. Subjects with AHY and hypercholesterolemia (HYCH), given their age, were included only if they were well compensated and with minimal medical therapy for no more than 3 years. Exclusion criteria included non-compliant subjects or those with ocular or systemic conditions such as keratoconjunctivitis, retinal detachment, glaucoma, uveitis, papilledema, ocular surgery, recent surgery in other systems, recent myocardial infarction, coronary syndrome, cerebral ischemia, renal or hepatic failure, respiratory infections, pulmonary embolism, recent pneumothorax, recent systemic trauma, aortic aneurysm, hemoptysis, hypertensive crisis, autoimmune diseases, etc.

After recruitment, everyone underwent an ocular examination including visual acuity assessment using the logarithm of the minimum angle of resolution (logMAR), anterior segment observation with biomicroscopy, IOP measurement with a Goldmann tonometer, fundoscopy using a binocular indirect ophthalmoscope, and OCTA examination. Following the ocular exam, forced spirometry and hematological tests, including total thrombus formation analysis system (T-TAS) evaluation, were performed [20,21,22].

### 2.2. Procedures

*Pulmonary function tests* were performed for each subject using a spirometer (Quark PFT, Cosmed, Pavona, Italy), following the recommendations of the American Thoracic Society and European Respiratory [23]. This method allows the diagnosis of conditions such as asthma and COPD [24]. Expiratory airflow is generally assessed by spirometry, with the following indices (Figure 1):Forced expiratory volume in the first second (FEV1).Forced vital capacity (FVC): the total volume of air exhaled during one forced breath.FEV1-to-FVC ratio helps differentiate between restrictive and obstructive disorders. The latter is defined by a reduced FEV1/FVC ratio below the 5th percentile of the predicted value.

Spirometry must be performed following precise execution protocols. Participants were instructed to wear comfortable clothing and to avoid intense physical activity, large meals, and the use of medications such as antihistamines, corticosteroids, and bronchodilators within 24 h prior to the test. The test was repeated three times due to its strong dependence on the patient’s level of cooperation. Age, gender, weight, height, and ethnicity were essential for selecting the appropriate predictive model in the spirometry device used [23].

*A Total thrombus formation analysis system (T-TAS)* (T-TAS^®^01, Fujimori Kogyo Co., Tokyo, Japan, distributed by GEPA S.r.l., Milan, Italy) was used to evaluate the impact of smoking and its relationship with thrombus formation. To perform the test, 400 µL of whole blood from 20 SMs and 10 non-SMs was collected into tubes containing benzylsulfonyl-D-argininyl-prolyl-4-amidinobenzylamide (BAPA). Subsequently, 340 µL of the sample was transferred into the PL-chip and analyzed. The growth, intensity, and stability of the formation of platelet clots were measured by the time needed to reach the occlusion pressure (OT), and the area under the flow–pressure curve (AUC) parameter, that is, an area under the pressure curve from the start of the test to a time of 10 min (Figure 1).

*The OCTA* (Optovue AngioVue^TM^ Fremont Inc., Fremont, CA, USA) allows for the detailed evaluation of the various layers of the retina and choroid and provides high-definition visualization of the vascular tree through interferometric measurement. This results in axial and transverse scans that, due to the circulatory movement in the vascular tree, show variations in light reflection and dispersion across different scans. Thus, OCTA is capable of detecting chorioretinal vascularization through the contrast generated by the movement of erythrocytes. Specifically, it is possible to observe a superficial vascular complex, a radial peripapillary superficial vascular complex, a deep vascular complex derived from the small vessels of the superficial plexus, and the foveal avascular zone (FAZ) devoid of capillaries. The remaining parameters examined in this study were macular vascular density (FD) and choriocapillary flow (Ch-Flow). These data were measured and calculated using automated algorithms provided by the device’s software.

### 2.3. Statistical Analysis

Descriptive statistics were obtained for all variables. Mean and ± standard deviation (SD) were used for continuous variables that were normally distributed, while the median and interquartile ranges were used for variables with asymmetric distribution. Categorical variables were presented as absolute numbers and proportions. A descriptive analysis of the parameters obtained at different sampling moments was performed. Comparisons between groups were assessed using an independent t-test or Mann–Whitney U test for continuous variables and Fisher’s exact test for categorical variables. This value represents an underestimation of the true unobserved OT. AUC was calculated for individuals who successfully reached OST and OT values. Differences in parameters between SMs and non-SMs were studied using mixed-effect models with group and time as covariates, and subject considered as a random effect. Statistical significance was set at *p* < 0.05. All statistical analyses were performed using the SPSS software (version 25.0; IBM Corp., Armonk, NY, USA).

## 3. Results

A total of 30 subjects were enrolled in this study, including 20 SMs and 10 non-SMs. The descriptive parameters and the mild comorbidities allowed by the inclusion criteria for the two groups, AHY and HYCH, are reported in Table 1 and Table 2.

After the eye examination, data were collected using OCTA regarding chorioretinal indicative parameters: superficial retinal vascular density (SVD), deep retinal vascular density (DVD), FAZ, FD, and Ch-Flow. The data from the right eye (RE) were used for statistical analysis. Table 3 shows the OCTA values obtained from the analysis of RE in both populations.

The two groups, after the eye examination, underwent pulmonary function tests, and the following indices of respiratory function were measured: FVC in liters and percentage, FEV1 in liters and percentage, and the FEV1/FVC ratio in percentage (Table 4).

Finally, the data obtained from the blood sample were analyzed for the evaluation of T-TAS and collected for the SMs group and the non-SMs group (Table 5).

### 3.1. Descriptive Parameters

A total of 30 eyes from 30 subjects recruited for the study were examined, divided into 20 REs from SMs and 10 REs from non-SMs. The SMs group included seven males (35%) and thirteen females (65%), with a mean age of 58.75 years (±8.30 SD). The average number of cigarettes smoked daily and the average years of smoking were 21.20 (±9.27 SD) and 39.10 (±8.56 SD), respectively, while the average pack years in the sample were 42.35 (±25.8 SD). Eight SMs (40%) were undergoing treatment for AHY, while HYCH was present in six SMs (30%). The mean values of the data extracted from the SMs group are reported in Table 6.

The non-SM group consisted of two males (20%) and eight females (80%), with a mean age of 55.70 years (±3.83 SD). Two subjects (20%) had AHY and were undergoing treatment, while two subjects (20%) had HYCH. Upon examination of the anterior segment and the fundus, none of the subjects exhibited noteworthy alterations. The mean values of the data extracted from the non-SM group are reported in Table 7.

### 3.2. Qualitative Parameters

The qualitative parameters considered were sex, AHY, and HYCH. Each parameter was analyzed by comparing the corresponding values between the two groups, and to evaluate the possible association between variables, Fisher’s exact test was used (Table 8). Upon reviewing the results and focusing on Fisher’s exact test, no statistically significant difference (*p* > 0.05) was found between the two populations for any of the qualitative parameters examined, including sex, AHY, and HYCH, with *p*-values of 0.675, 0.419, and 0.682, respectively.

### 3.3. Quantitative Parameters

The quantitative parameters observed were IOP, logMAR, SVD, DVD, FAZ, FD, Ch-FLOW, FVC L, FVC %, FEV1 L, FEV1 %, FEV1/FVC ratio %, T-TAS OT, and T-TAS AUC. To evaluate potential differences between the two groups, the Mann–Whitney U test for independent samples was performed. A non-parametric test was used due to the non-normality of the distributions and the small sample size.

Evaluating the values of IOP and visual acuity (logMAR) in RE, no statistically significant differences were found between the examined populations: *p*-values of 0.78 and 0.53, respectively (Figure 2). The same result was found in the OCTA examination (Figure 3) for the parameters SVD and DVD, with *p*-values of 0.11 and 0.78, respectively. Continuing the data analysis with OCTA, other parameters were assessed: the FAZ and Ch-Flow of RE, with *p*-values of 0.619 and 0.559, respectively, showing no statistical significance. In contrast, the FD of RE was statistically significant, with a *p*-value of 0.049.

As far as the pulmonary function is concerned, the difference in FEV1 L among the two groups resulted as statistically significant with a *p*-value of 0.019. Instead, no statistically significant difference was found in the other values, as the *p*-values for FVC L, FVC %, FEV1 %, and the FEV1/FVC ratio were calculated as 0.094, 0.267, 0.214, and 0.779, respectively (Figure 4).

Thanks to an extremely low *p*-value of 0.0000 for both T-TAS OT and T-TAS AUC, it was possible to assert the presence of a statistically significant difference between the two groups examined, testifying the pathological process of primary hemostasis (Figure 5).

## 4. Discussion

Based on results from the scientific literature, we initiated investigations to collect data on respiratory functional characteristics, critical thrombus formation values, and flowmetry data of the chorioretinal vascular district in subjects participating in the SMs and non-SMs study groups. In this context, T-TAS values revealed a highly significant difference between the groups, highlighting alterations in primary hemostasis as assessed by this novel technology. Specifically, accelerated thrombus formation was observed in the SMs group compared to the non-SMs group, as indicated by a reduced T-TAS OT and increased T-TAS AUC in SMs. The same pattern was observed in spirometric measures, with reduced lung function indicated by a significant difference in FEV1, and in chorioretinal vascular flow, as evidenced by a reduction in FD in the examined groups.

Our research was motivated by numerous studies confirming that tobacco products, when consumed over extended periods, are associated with the development of complex diseases that have both acute and chronic harmful effects. These effects influence oxidative stress, antioxidant metabolites, coagulation factors, vascular flow, overall blood circulation, and direct factors promoting carcinogenesis. Smoking accelerates atherosclerosis and related events, including myocardial infarction, stroke, and various arteriosclerotic diseases [25,26,27,28].

One of the most significant stimuli for aging in the human body is constant exposure to environmental factors, which, in combination with epigenetic characteristics and genetic background, can modulate individual responses, including those related to the harmful effects of tobacco smoke inhalation. These events contribute to increased cardiovascular risk, leading to HYCH, platelet aggregation, atherosclerosis, atherothrombotic events, peripheral thrombosis, and metabolic diseases [29,30,31,32,33].

Furthermore, it is widely recognized that age-related macular degeneration (AMD), in its three specific forms—colloid, atrophic, and exudative—has cigarette smoke exposure as its primary etiopathogenetic cause. The study by Yang et al. reports investigations performed using OCTA on four groups of subjects: SMs and non-SMs, with and without AMD. The research highlighted a higher vascular density in non-SMs and a negative bivariate correlation between deep vascular density and the pack/year history in SMs. In addition, central retinal thickness was significantly higher in non-SM subjects with dry AMD compared to SMs [34].

In another study, blood pressure and heart rate were considered and were significantly higher in SMs immediately after smoking, while peripapillary and parafoveal vessel density and mean avascular zone area on OCTA did not change, as expected [18].

The study by Eriş et al. reported that central choroidal thickness, examined with OCTA, was reduced in SMs, while macular vascular perfusion and optic disk perfusion were not statistically significant. The smoking period in the study was at least 5 years with a pack of cigarettes per day. They concluded that further prospective studies are needed to clarify the relationship between smoking and choroidal vascular disease [35].

Dogan et al. reported that macular vascular densities observed with OCTA were significantly lower in the SM group compared to the control group. The mean exposure to cigarette smoking was 3.3 ± 1.0 pack/years. According to the authors, retinal vascular changes caused by cigarette smoking may occur even with low pack/year exposure. These changes, which can be demonstrated by OCTA, may reflect the early impact of cigarette smoking on the microvascular system [17].

Other data regarding OCTA examination in 45 SMs, who had smoked for an average of 2.2 years at 20 cigarettes per day, compared to non-SMs, highlighted significant variations between the two groups in terms of various retinal thicknesses and vascular flow. In conclusion, the authors noted that despite the short-term smoking duration, ischemic effects were observed in the retinochoroidal and vascular structures [19].

The findings from various authors described above partly agree with our results from the OCTA examination. Furthermore, our research has expanded upon existing data not only regarding the ocular vascular district but also by evaluating respiratory function and basic factors involved in thrombus formation and atherosclerosis. In our study, we included long-term smokers who had smoked for an average of 39.10 years (±8.56 SD), consumed an average of 21.20 cigarettes (±9.27 SD) per day, and had a pack/year history of 42.35 (±25.00). This approach allowed us to assess chronic damage, in contrast to other studies, and we observed significant alterations in the hematological data, indicating important systemic damage.

The substantial body of literature confirms that cigarette smoking directly influences atherothrombosis, promoting platelet activation, adhesion, aggregation, and the coagulation factor cascade while altering the balance of fibrinolysis. The primary mechanism appears to be endothelial dysfunction, characterized by reduced NO availability and increased superoxide anion production due to smoke inhalation. Subsequently, a chronic inflammatory response is activated in the vascular walls, involving endothelial cells, macrophages, and dendritic cells [36,37].

This response is mediated by receptors that play a role in immune system function, such as TLRs, and by molecular structures released by damaged or necrotic cells, alongside the activation of matrix metalloproteinases (MMPs), which contribute to vulnerability to atherothrombotic events. Analyzing the results of our study, particularly those concerning T-TAS and FD values, we hypothesize a pathogenetic sequence that differs from what has been previously reported in the literature. The T-TAS OT is significantly reduced, and thrombus stability, T-TAS AUC, is significantly increased in SMs compared to the control group. Similarly, a region with high energetic and metabolic demands for fine visual functions such as the macula is represented by a statistically reduced FD in SMs. This suggests that the primary cause of circulatory alterations in individuals dependent on cigarette smoking may not always be strictly linked to endothelial damage. Conversely, endoluminal biochemical events, characterized by reduced flow and/or concomitant platelet aggregation, could act as both primary and secondary triggers depending on the atherothrombotic events.

However, the specific biomolecular events that activate local structural and vascular changes remain unknown. It is likely that a combination of factors—such as an imbalance in the coagulation system, redox system dysregulation, cellular senescence, impaired cell turnover due to autophagy dysfunction, and individual genetic and epigenetic backgrounds—are responsible for the degenerative development not only of the chorioretinal system but also of the cardiovascular, pulmonary, and neurological systems in individuals with tobacco dependence [38,39].

This suggests that smoking cessation is a challenging but dynamic process, and all SMs would undoubtedly benefit from cessation, even after the development of significant cardiovascular diseases [40].

### Limitations and Future Study Strategies

Since the addiction to tobacco smoking has strong roots in SMs, few people turn to the designated clinical centers. Thus, our study has as its main limitation the number of subjects examined. Additionally, the research focused only on the impact of traditional cigarette smoking, while other modalities, such as the use of electronic cigarettes or non-tobacco-based products, were not evaluated. The sample size of the control group and the gender-based recruitment should be improved and better balanced in the next study to enhance the robustness of the statistical analysis results. Another limitation was the relatively high average age of the participants, as long-term SMs typically seek assistance at our smoking cessation center only when motivated to quit, which is less common among younger individuals.

Moreover, the avenue for further study will be to examine and correlate additional spirometric parameters, preferably those with higher sensitivity to smoking-induced pulmonary damage, about coagulation and oxidative stress data, flowmetry, and chorioretinal morphology before and after smoking cessation, with a careful follow-up to explore possible differences in parameters. Finally, another very interesting research direction will be to assess SMs with AMD, of the colloid, atrophic, or wet type, compared to non-SMs. It is now well established that these retinal pathologies, often debilitating due to severe visual deficits, develop in overt and chronic situations, such as tobacco dependence.

Therefore, equivalent nicotine consumption is associated with multidimensional adverse effects on a wide range of biological and physiological markers that are crucial for maintaining good health.

## 5. Conclusions

This study demonstrated the potential impact of cigarette smoking on retinal vascular parameters, highlighting alterations in primary hemostasis markers in SMs compared to non-SMs, in addition to the well-established effects on respiratory function.

In summary, the damage caused by smoking manifests as functional disturbances and variations across multiple systems in the human body, including the respiratory, circulatory, and visual systems, all of which were explored in the present study.

Considering this, it is essential to continue the recruitment of subjects and expand the dataset to evaluate and confirm whether vascular density in the macula lutea, in conjunction with other angiotomographic data, can be considered a suitable parameter for exploring the damage caused by cigarette smoke or other equivalent devices.

Given the importance of the topic for public health, and the socio-economic consequences of smoking-related damage, it is hoped that the scientific literature will continue to expand through multidisciplinary investigations conducted by experts in the field.

Further research will be crucial to clarify the impact of smoking exposure and its downstream effects, particularly about the duration and dose of exposure to the inhaled toxic substances.

## Figures and Tables

**Figure 1 medicina-61-00347-f001:**
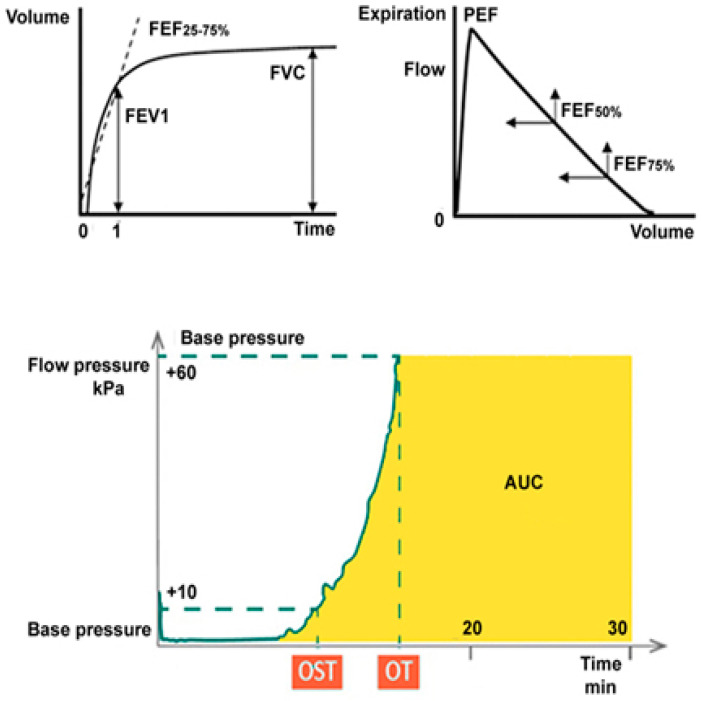
Spirometric curves (**upper**). Volume–time curve (seconds) and flow–volume curve. Forced expiratory flow values in percentage (FEF); the volume of air exhaled during the first second of a forced exhalation (FEV1); forced vital capacity, or the total volume the patient exhales during a maximal exhalation (FVC); peak expiratory flow rate (PEF). Parameters of the total thrombus formation analysis system (T-TAS) (**under**): onset of occlusion time (OST); occlusion time (OT).

**Figure 2 medicina-61-00347-f002:**
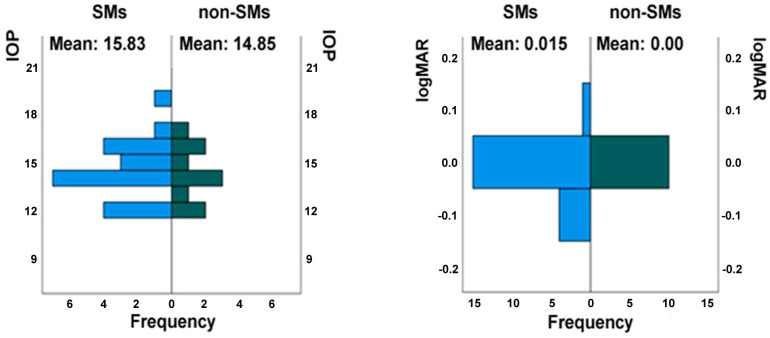
Mann–Whitney U test for independent samples of 20 smokers (SMs) and 10 non-SMs in the right eye (RE): intraocular pressure (IOP) in mmHg (**left panel**); visual acuity in logMAR (**right panel**).

**Figure 3 medicina-61-00347-f003:**
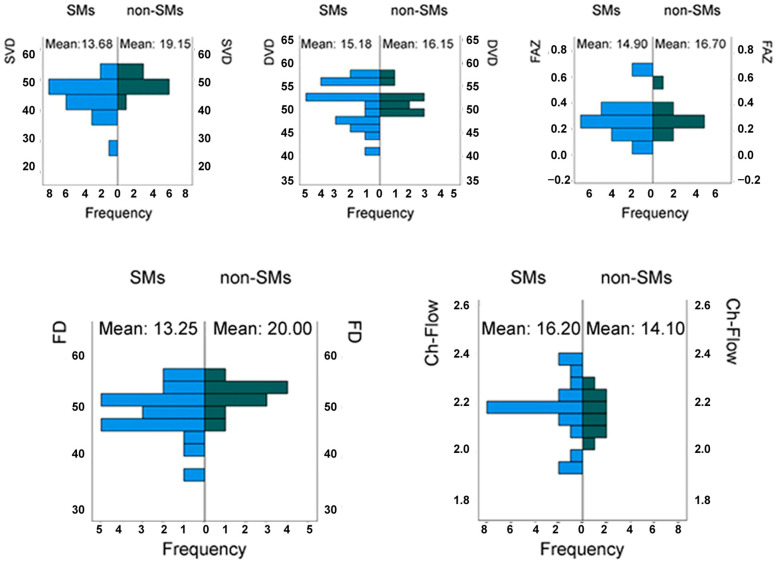
Mann–Whitney U test for independent samples of 20 smokers (SMs) and 10 non-SMs in the right eye (RE): superficial vascular density (SVD) in %; deep vascular density (DVD) in %; foveal avascular zone (FAZ) in %; macular vascular density (FD) in %; choriocapillary flow (Ch-Flow) in %.

**Figure 4 medicina-61-00347-f004:**
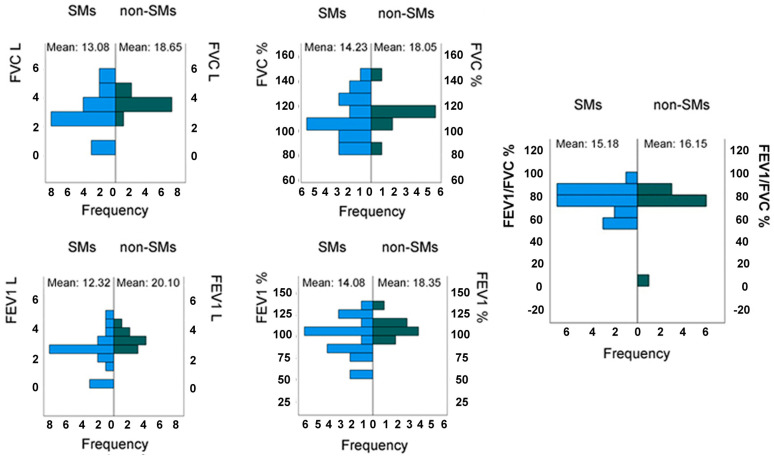
Mann–Whitney U test for independent samples of 20 smokers (SMs) and 10 non-SMs: total volume during maximal exhalation in liters (FVC L) and percentage (FVC %); volume of air exhaled during the first second of a forced exhalation in liters (FEV1 L) and percentage (FEV1 %); FEV1/FVC ratio in %, i.e., the ratio of the volume of air exhaled during the first second of a forced exhalation (FEV1) to the total volume during maximal exhalation (FVC).

**Figure 5 medicina-61-00347-f005:**
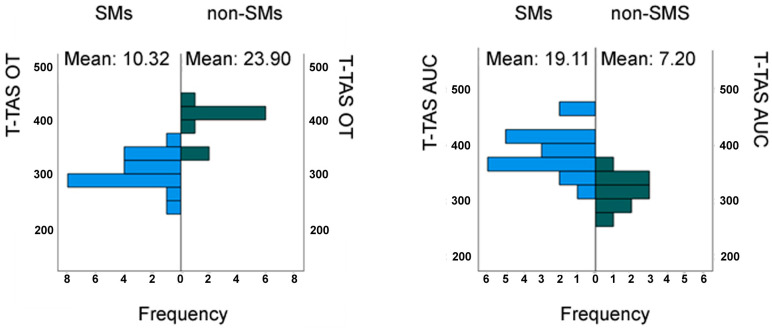
Mann–Whitney U test for independent samples of 20 smokers (SMs) and 10 non-SMs: total thrombus formation analysis system (T-TAS), occlusion time (OT), and area under the curve (AUC) quantifying thrombus stability expressed in min × KPa.

**Table 1 medicina-61-00347-t001:** Descriptive parameters and comorbidities of smokers (SMs).

SM	Sex	Age (Years)	Cigarettes/Day	Years of Smoking	Packages/Year	AHY	HYCH
1	F	59	20	40	40	0	HYCH
2	F	62	20	40	40	AHY	HYCH
3	M	61	20	44	44	0	HYCH
4	F	60	10	40	20	0	0
5	M	61	15	40	30	0	0
6	M	40	20	27	27	0	0
7	M	70	18	55	49.5	0	0
8	M	67	20	25	25	AHY	0
9	F	64	20	45	45	0	0
10	M	70	30	50	75	AHY	0
11	F	64	50	50	125	AHY	0
12	F	70	10	45	22.5	AHY	0
13	F	61	20	40	40	0	HYCH
14	F	52	18	38	34.2	0	0
15	F	49	20	33	33	0	0
16	M	48	18	33	29.7	0	HYCH
17	F	59	20	42	42	AHY	0
18	F	50	15	36	27	AHY	0
19	F	49	20	20	20	0	0
20	F	59	40	39	78	AHY	HYCH

Arterial hypertension (AHY); hypercholesterolemia (HYCH).

**Table 2 medicina-61-00347-t002:** Descriptive parameters and comorbidities of non-smokers (non-SMs).

Non-SM	Sex	Age (Years)	AHY	HYCH
1	F	57	0	0
2	F	60	0	0
3	F	55	0	0
4	F	50	AHY	0
5	F	50	0	0
6	M	53	AHY	0
7	F	61	0	HYCH
8	F	56	0	0
9	F	56	0	0
10	M	59	0	HYCH

Arterial hypertension (AHY); hypercholesterolemia (HYCH).

**Table 3 medicina-61-00347-t003:** Parameters of optical coherence tomography angiography (OCTA) in the right eye (RE) of smokers (SMs) and non-smokers (non-SMs).

SM	SVD (%)	DVD (%)	FAZ (mm^2^)	FD (%)	Ch-Flow (%)
1	46.5	50.3	0.060	42.02	2.296
2	49.2	53.2	0.263	49.77	2.188
3	42.7	48.4	0.207	48.24	2.186
4	36.8	46.7	0.350	47.29	2.363
5	44.5	55.6	0.149	49.29	2.182
6	47.7	57.0	0.203	54.47	2.162
7	29.6	53.1	0.619	36.71	2.107
8	51.9	56.3	0.065	46.53	2.050
9	48.3	45.8	0.235	51.07	2.107
10	38.2	53.1	0.217	45.89	1.957
11	40.9	46.7	0.629	56.36	2.195
12	46.8	56.0	0.197	53.00	2.159
13	43.3	44.5	0.365	47.30	2.190
14	38.7	46.5	0.320	44.18	1.946
15	48.7	57.8	0.151	51.40	2.218
16	43.0	41.3	0.221	50.38	2.369
17	42.8	47.8	0.307	46.84	1.949
18	48.0	52.3	0.264	50.36	2.173
19	47.9	53.0	0.309	56.80	2.325
20	52.2	55.8	0.138	50.90	2.224
**Non-SM**	**SVD (%)**	**DVD (%)**	**FAZ (mm^2^)**	**FD (%)**	**Ch-Flow (%)**
1	46.7	50.3	0.213	52.25	2.236
2	46.2	56.2	0.286	54.11	2.121
3	48.7	53.0	0.256	49.46	2.057
4	48.0	50.3	0.274	54.03	2.157
5	48.0	58.1	0.172	51.25	2.067
6	50.0	49.4	0.292	52.19	2.238
7	46.5	52.0	0.532	55.17	2.034
8	51.9	51.8	0.323	52.88	2.147
9	41.9	49.0	0.133	45.13	2.257
10	50.0	48.6	0.306	53.73	2.179

Superficial retinal vascular density (SVD), and deep retinal vascular density (DVD); foveal avascular zone (FAZ); macular vascular density (FD) and flow (Ch-Flow).

**Table 4 medicina-61-00347-t004:** Spirometry results in smokers (SMs) and non-SMs.

SM	FVC L	FVC %	FEV1 L	FEV1 %	FEV1/FVC Ratio %
1	2.78	104.0	2.28	101.0	82.0
2	2.36	98.0	2.02	101.0	85.8
3	5.22	121.0	3.93	116.0	75.0
4	3.30	139.0	2.53	127.0	76.7
5	6.11	138.0	4.18	128.0	73.2
6	4.89	101.0	4.16	105.0	85.0
7	4.21	108.0	2.39	80.0	56.7
8	5.26	123.0	3.41	103.0	65.0
9	2.88	116.0	1.81	87.0	62.9
10	3.36	91.0	1.68	59.0	50.0
11	2.12	82.0	1.16	54.0	54.7
12	3.13	142.0	2.40	133.0	76.8
13	2.77	106.0	2.02	92.0	94.0
14	2.76	102.0	2.03	89.0	73.6
15	3.26	112.0	2.72	109.0	83.0
16	6.01	127.0	5.12	124.0	78.0
17	2.58	80.0	2.10	76.0	81.0
18	3.84	92.2	2.70	78.0	70.31
19	2.89	87.0	2.09	83.0	81.3
20	2.83	105.0	2.30	101.0	81.0
**Non-SM**	**FVC L**	**FVC %**	**FEV1 L**	**FEV1 %**	**FVE1/FVC Ratio %**
1	3.66	115	2.60	96	71.1
2	3.07	116	2.26	102	73.6
3	3.61	115	2.79	105	77.7
4	3.01	109	2.44	104	81.1
5	3.35	118	2.66	110	79.4
6	3.80	86	3.21	91	84.6
7	2.78	109	2.44	114	87.8
8	3.60	115	2.68	100	74.4
9	4.59	148	3.45	131	75.2
10	4.40	114	3.50	114	79.5

Total volume during maximal exhalation (FVC) in liters and percentage; volume of air exhaled during the first second of forced exhalation (FEV1) in liters and percentage; FEV1/FVC ratio in percentage.

**Table 5 medicina-61-00347-t005:** Hemorheological parameters in smokers (SMs) and non-smokers (non-SMs).

SM	T-TAS OT (s)	T-TAS AUC (min *×* Kpa)
1	304	384.5
2	286	452.0
3	285	409.0
4	360	311.6
5	342	367.4
6	291	370.8
7	342	368.9
8	298	393.9
9	329	372.4
10	334	345.1
11	264	411.9
12	286	452.0
13	231	410.8
14	324	365.2
15	290	368.9
16	288	423.2
17	322	325.2
18	296	416.9
19	319	376.9
20	305	385.6
**Non-SM**	**T-TAS OT (s)**	**T-TAS AUC (Min *×* Kpa)**
1	401	316.1
2	426	290.3
3	410	301.2
4	327	346.8
5	413	320.2
6	381	342.2
7	344	370.9
8	414	255.1
9	403	330.3
10	409	286.2

Total thrombus formation analysis system: T-TAS; occlusion time: OT; area under the curve (AUC) quantifying thrombus stability.

**Table 6 medicina-61-00347-t006:** Descriptive parameters in the smokers (SMs) group.

SMs DATA	MEAN	±SD
Age (years)	58.75	8.30
Number of cigarettes	21.20	9.27
Years of smoking	39.10	8.56
Packages/year	42.35	25.00
IOP RE (mmHg)	14.55	1.82
Visual acuity RE (logMAR)	−0.015	0.0489
SVD (%) RE	44.39	5.57
DVD (%) RE	51.06	4.80
FAZ (mm^2^) RE	0.26	0.15
FD (%) RE	48.94	4.75
Ch-Flow (%) RE	2.17	0.12
FVC L	2.89	1.50
FVC %	108.71	18.49
FEV1 L	2.09	1.12
FEV1 %	97.30	22.13
FEV1/FVC ratio %	74.30	11.36
T-TAS OT (s)	304.79	30.86
T-TAS AUC (min × KPa)	385.61	37.78

Intraocular pressure (IOP); right eye (RE); assessment of visual acuity in logarithm of the minimum angle of resolution (logMAR); superficial retinal vascular density (SVD) and deep retinal vascular density (DVD); foveal avascular zone (FAZ); macular vascular density (FD); choriocapillary flow (Ch-Flow); total volume during maximal exhalation in liters (FVC L) and percentage (FVC %); volume of air exhaled during the first second of forced exhalation in liters (FEV1 L) and percentage (FEV1 %); total thrombus formation analysis system (T-TAS); occlusion time (OT); area under the curve (AUC) quantifying thrombus stability.

**Table 7 medicina-61-00347-t007:** Descriptive parameters in the non-smoker (non-SM) group.

Non-SMs DATA	Mean	±SD
Age (years)	55.70	3.83
IOP (mmHg) RE	14.30	1.70
Visual acuity (logMAR) RE	0.000	0.0000
SVD (%) RE	47.79	2.74
DVD (%) RE	51.87	3.14
FAZ (mm^2^) RE	0.28	0.1078
FD (%) RE	52.02	2.92
Ch-Flow (%) RE	2.15	0.08
FVC L	3.59	0.58
FVC %	114.50	14.95
FEV1 L	2.80	0.44
FEV1%	106.70	11.29
FEV1/FVC ratio %	71.64	23.19
T-TAS OT (s)	392.80	32.58
T-TAS AUC (min × KPa)	315.93	33.91

Intraocular pressure (IOP); right eye (RE); assessment of visual acuity in logarithm of the minimum angle of resolution (logMAR); superficial retinal vascular density (SVD) and deep retinal vascular density (DVD); foveal avascular zone (FAZ); macular vascular density (FD); choriocapillary flow (Ch-Flow); total volume during maximal exhalation in liters (FVC L) and percentage (FVC %); volume of air exhaled during the first second of forced exhalation in liters (FEV1 L) and percentage (FEV1 %); total thrombus formation analysis system (T-TAS); occlusion time (OT); area under the curve (AUC) quantifying thrombus stability.

**Table 8 medicina-61-00347-t008:** Fisher’s exact test between the two groups of smokers (SMs) and non-smokers (non-SMs) to evaluate the possible association between variables: sex and comorbidities.

		SMs	Non-SMs	Total
0	SEX COUNT	7	2	9
	% Sex	77.8%	22.2%	100.0%
	% in group	35.0%	20.0%	30.0%
	% of total	23.3%	6.7%	30.0%
1	SEX COUNT	13	8	21
	% Sex	61.9%	38.1%	100.0%
	% in group	65.0%	80.0%	70.0%
	% of total	43.3%	26.7%	70.0%
Total	SEX COUNT	20	10	30
	% Sex	66.7%	33.3%	100.0%
	% in group	100.0%	100.0%	100.0%
	% of total	66.7%	33.3%	100.0%
0	AHY COUNT	12	8	20
	% AHY	60.0%	40.0%	100.0%
	% in group	60.0%	80.0%	66.7%
	% of total	40.0%	26.7%	66.7%
1	AHY COUNT	8	2	10
	% AHY	80.0%	20.0%	100.0%
	% in group	40.0%	20.0%	33.3%
	% of total	26.7%	6.7%	33.3%
Total	AHY COUNT	20	10	30
	% AHY	66.7%	33.3%	100.0%
	% in group	100.0%	100.0%	100.0%
	% of total	66.7%	33.3%	100.0%
0	HYCH COUNT	14	8	22
	% HYCH	63.6%	36.4%	100.0%
	% in group	70.0%	80.0%	73.3%
	% of total	46.7%	26.7%	73.3%
1	HYCH COUNT	6	2	8
	% HYCH	75.0%	25.0%	100.0%
	% in group	30.0%	20.0%	26.7%
	% of total	20.0%	6.7%	26.7%
Total	HYCH COUNT	20	10	30
	% HYCH	66.7%	33.3%	100.0%
	% in group	100.0%	100.0%	100.0%
	% of total	66.7%	33.3%	100.0%

Arterial hypertension (AHY); hypercholesterolemia (HYCH).

## Data Availability

Data are available if requested from the corresponding author.

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
