# Peer review of "Multidisciplinary Clinical Study on Retinal, Circulatory, and Respiratory Damage in Smoking-Dependent Subjects"

_medicina, 2025, doi:10.3390/medicina61020347_

Round 1
Reviewer 1 Report
Comments and Suggestions for Authors
• A brief summary (one short paragraph) The aim of the paper is to describe the damages that occur in
smokers compared to non smoker at the level of the retina, circulation and respiratory system.
• General concept comments
Article: Weakness:
1-The article uses 2:1 ratio for cases and controls which weakens the results(not mentioned in the
limitations section)
2-The ratio of male to female in the sample is skewed with significantly more females in both controls and
cases (also not mentioned in the limitations section).
3-There is no mention of the demographics of the subjects, race and ethnicity which could be an important
determinant of the study.
4-Sample size is small, it would be much more powerful if the sample size is increased.
Review: No mention of any references related to OCTA analysis in smokers vs. non smokers.- Need to
add all the OCTA references to the article to make it sound scientific and supported and cite different
articles in the introduction and in the discussion section to compare and contrast your findings to the
literature.
(https://pubmed.ncbi.nlm.nih.gov/?term=retinal+vascular+changes+using+OCTA+in+smokers). Currently
authors state in the conclusion: "Considering the lack of literature on the relationship between smoking
dependency risk factors and their effect on chorioretinal circulation, further investigation of this finding will
be necessary to expand the value of the results obtained." This statement is inaccurate, it needs to be
modified after citing the OCTA papers that clearly demonstrate the status of the chorioretinal circulation
differences and impudence in smokers compared to non smokers.
• Specific comments Figure 2 right panel, please review the log Mar values used (14.75 and 17
respectively), Those do not look like they are Log Mar val
Author Response
All Co-authors thank the 3 Reviewers who helped identify the changes needed to improve the manuscript and, in case it is accepted by the journal, allow the dissemination of the scientific results.
Author's Reply to the Review Report (Reviewer 1)
- General concept comments
Article: Weakness:
1-The article uses 2:1 ratio for cases and controls which weakens the results(not mentioned in the
limitations section).
2-The ratio of male to female in the sample is skewed with significantly more females in both controls and
cases (also not mentioned in the limitations section).
- Reviewer 1 correctly suggested emphasizing the numerical difference between the two groups in the 'Limitations' section of the study. We encoutered difficulties in recruiting healthy non-smoker subjects and were limited to 10 individuals. We plan to continue recruiting additional controls for future investigations. Additionally, we will ensure a balanced distribution of subjects between females and males (see Limitations Section).
3-There is no mention of the demographics of the subjects, race and ethnicity which could be an important
determinant of the study.
- Thirty Caucasian Italian subjects were enrolled in the study (see Methods section).
4-Sample size is small, it would be much more powerful if the sample size is increased.
- The critique made by the Reviewer is valid, and we have acknowledged this as a limitation of the research. Subjects with tobacco addiction are rarely convinced to quit smoking, and even less likely to undergo examination. Our work can be considered a pilot study, and we plan to continue recruiting both smokers and non-smokers to further investigate the issues described in the manuscript.
Review: No mention of any references related to OCTA analysis in smokers vs. non smokers.- Need to
add all the OCTA references to the article to make it sound scientific and supported and cite different
articles in the introduction and in the discussion section to compare and contrast your findings to the
literature.
(https://pubmed.ncbi.nlm.nih.gov/?term=retinal+vascular+changes+using+OCTA+in+smokers). Currently
authors state in the conclusion: "Considering the lack of literature on the relationship between smoking
dependency risk factors and their effect on chorioretinal circulation, further investigation of this finding will
be necessary to expand the value of the results obtained." This statement is inaccurate, it needs to be
modified after citing the OCTA papers that clearly demonstrate the status of the chorioretinal circulation
differences and impudence in smokers compared to non smokers.
- The sentence has been modified (see page 17).
- References have been reviewed and added to the Introduction and Discussion sections.
- Specific commentsFigure 2 right panel, please review the log Mar values used (14.75 and 17
respectively), Those do not look like they are Log Mar val
- We thank the Reviewer for the comment and have corrected the error in Figure 2 accordingly.

Reviewer 2 Report
Comments and Suggestions for Authors
Comprehensive analysis of the smoking dependency with taking into consideration of the whole body is valuable. That's why I would like to congratulate.
Author Response
All Co-authors thank the 3 Reviewers who helped identify the changes needed to improve the manuscript and, in case it is accepted by the journal, allow the dissemination of the scientific results.
Author's Reply to the Review Report (Reviewer 2)
Comprehensive analysis of the smoking dependency with taking into consideration of the whole body is valuable. That's why I would like to congratulate.
- We sincerely thank Reviewer 3 for congratulating us on our research.

Reviewer 3 Report
Comments and Suggestions for Authors
First and foremost, I'd like to congratulate the authors on their multidisciplinary clinical study, which investigated changes in hemorheological and respiratory characteristics, as well as chorioretinal damage, in 20 right eyes of 20 SMs and compared them to 10 non-SMs. The study was motivated by the fact that traditional cigarette smoking can increase oxidative stress, platelet activation, and thrombotic phenomena at the systemic level, and that fundoscopy can provide information about the cerebral district and the entire circulatory system. In this regard, they investigated the effects of cigarette smoking on chorioretinal vascularization, as well as pulmonary and blood parameters, using OCTA, spirometry, and the total thrombus formation analysis system (T-TAS). Their findings supported the systemic influence of smoking, as demonstrated by an increase in T-TAS and a decrease in forced expiratory volume in 1 second expressed in liters (FEV1 L) in SMs as compared to the non-SM group. Furthermore, there was a considerable change in macular vascular density (FD) in SMs. The authors therefore concluded that FD, FEV1, and T-TAS could be useful parameters for linking SM changes. This was substantiated by changes found in OCTA, spirometry, and hemorheological data. Yet, the authors went on to highlight that more studies could be of benefit to confirm and broaden the results already obtained and to analyze the systemic vascular damage and oxidative stress induced by tobacco usage.
As noted by the authors, despite the fact that smoking is clearly hazardous to one's health, millions of people continue to engage in this chronic and obsessive behavior, which has long been linked to the emergence of a variety of diseases. Furthermore, it appears that this multidisciplinary study was conducted in such a way that the step-by-step evaluation of the entire examination procedure was entirely transparent and thorough, despite the fact that the number of study participants was rather small. Indeed, after reviewing the literature, it is clear that there is a need for more research publications on this seemingly hot topic, taking into account the effects of smoking from various perspectives.
Overall, with a few minor issues rectified, I feel this study has the potential to make a large, if not massive, positive contribution to the worldwide literature, given the global aging population and the prevalence of smoking, as well as the related health and economic burden.
MINOR ISSUES
o Line 52-113: The introduction is lengthy, complicated, and difficult to follow. The opening should be brief, with only a few phrases stating the aim. Otherwise, the study appears to be a review of the literature in its current form, which genuinely undermines readers' focus.
o Line 124: According to the results section, only the participant's right eye was evaluated. Please provide information about the approach you used to decide which eye to analyze.
o The phrase in Lines 186-188 may be more descriptive if placed after 'The SMs were required to…..'.
o In line 157-158 the authors mentioned that ‘Spirometry must be performed according to precise execution protocols, including assessing the subject's height and weight during the examination’. Nevertheless, no numerical data were provided in the findings section to indicate the actual values they intended to measure and their subsequent analysis.
Line 410-414: It appears that this paragraph would be better suited to the concluding section. Please double-check and correct any errors accordingly.
Author Response
All Co-authors thank the 3 Reviewers who helped identify the changes needed to improve the manuscript and, in case it is accepted by the journal, allow the dissemination of the scientific results.
Author's Reply to the Review Report (Reviewer 3)
First and foremost, I'd like to congratulate the authors on their multidisciplinary clinical study, which investigated changes in hemorheological and respiratory characteristics, as well as chorioretinal damage, in 20 right eyes of 20 SMs and compared them to 10 non-SMs. The study was motivated by the fact that traditional cigarette smoking can increase oxidative stress, platelet activation, and thrombotic phenomena at the systemic level, and that fundoscopy can provide information about the cerebral district and the entire circulatory system. In this regard, they investigated the effects of cigarette smoking on chorioretinal vascularization, as well as pulmonary and blood parameters, using OCTA, spirometry, and the total thrombus formation analysis system (T-TAS). Their findings supported the systemic influence of smoking, as demonstrated by an increase in T-TAS and a decrease in forced expiratory volume in 1 second expressed in liters (FEV1 L) in SMs as compared to the non-SM group. Furthermore, there was a considerable change in macular vascular density (FD) in SMs. The authors therefore concluded that FD, FEV1, and T-TAS could be useful parameters for linking SM changes. This was substantiated by changes found in OCTA, spirometry, and hemorheological data. Yet, the authors went on to highlight that more studies could be of benefit to confirm and broaden the results already obtained and to analyze the systemic vascular damage and oxidative stress induced by tobacco usage.
As noted by the authors, despite the fact that smoking is clearly hazardous to one's health, millions of people continue to engage in this chronic and obsessive behavior, which has long been linked to the emergence of a variety of diseases. Furthermore, it appears that this multidisciplinary study was conducted in such a way that the step-by-step evaluation of the entire examination procedure was entirely transparent and thorough, despite the fact that the number of study participants was rather small. Indeed, after reviewing the literature, it is clear that there is a need for more research publications on this seemingly hot topic, taking into account the effects of smoking from various perspectives.
Overall, with a few minor issues rectified, I feel this study has the potential to make a large, if not massive, positive contribution to the worldwide literature, given the global aging population and the prevalence of smoking, as well as the related health and economic burden.
MINOR ISSUES
o Line 52-113: The introduction is lengthy, complicated, and difficult to follow. The opening should be brief, with only a few phrases stating the aim. Otherwise, the study appears to be a review of the literature in its current form, which genuinely undermines readers' focus.
- We thank Reviewer 3 for the kind words regarding our research and have addressed the comments raised in the manuscript.
- The Introduction section has been shortened from its original form; however, we have included the sentences requested by Reviewer 1 concerning the references in the literature on smoking and OCT.
o Line 124: According to the results section, only the participant's right eye was evaluated. Please provide information about the approach you used to decide which eye to analyze.
- We are aware that, for statistical analysis, it is appropriate to report the values from only one eye, not both. Therefore, we selected the right eye for all the subjects examined and have included this information in the Results section.
o The phrase in Lines 186-188 may be more descriptive if placed after 'The SMs were required to…..'.
The sentence has been moved to line 130, as requested by Reviewer 3.
o In line 157-158 the authors mentioned that ‘Spirometry must be performed according to precise execution protocols, including assessing the subject's height and weight during the examination’. Nevertheless, no numerical data were provided in the findings section to indicate the actual values they intended to measure and their subsequent analysis.
- The data analysis is performed by the predictive model integrated into the device, which provides results tailored to the subject being examined.
- We have improved the sentence and included Ref.23 (recommendations from the American Thoracic Society and European Respiratory) “Spirometry must be performed according to precise execution protocols. Participants were instructed to wear comfortable clothing and avoid intense physical activity, large meals, and medications such as antihistamines, corticosteroids, and bronchodilators within 24 hours prior to the test. The test was repeated three times due to its strong dependence on the patient's level of cooperation. Age, gender, weight, height, and ethnicity were essential for selecting the predictive model in the spirometry device used.”
Line 410-414: It appears that this paragraph would be better suited to the concluding section. Please double-check and correct any errors accordingly.
- The note from the Reviewer was very useful; we have moved the sentence to the Conclusions section and reordered the existing sentences accordingly.
